# A Strategy for Gene Knockdown in Dinoflagellates

**DOI:** 10.3390/microorganisms10061131

**Published:** 2022-05-31

**Authors:** Miranda Judd, Allen R. Place

**Affiliations:** Institute of Marine and Environmental Technologies, University of Maryland Center for Environmental Science, 701 E Pratt St., Baltimore, MD 21022, USA; place@umces.edu

**Keywords:** dinoflagellate, knockdown, morpholino, translation

## Abstract

Dinoflagellates are unicellular protists that display unusual nuclear features such as large genomes, condensed chromosomes and multiple gene copies organized as tandem gene arrays. Genetic regulation is believed to be controlled at the translational rather than transcriptional level. An important player in this process is initiation factor eIF4E which binds the 7-methylguanosine cap structure (m7G) at the 5′-end of mRNA. Transcriptome analysis of eleven dinoflagellate species has established that each species encodes between eight to fifteen eIF4E family members. Determining the role of eIF4E family members in gene expression requires a method of knocking down their expression. In other eukaryotes this can be accomplished using translational blocking morpholinos that bind to complementary strands of RNA, therefore inhibiting the mRNA processing. Previously, unmodified morpholinos lacked the ability to pass through cell membranes, however peptide-based reagents have been used to deliver substances into the cytosol of cells by an endocytosis-mediated process without damaging the cell membrane. We have successfully delivered fluorescently-tagged morpholinos to the cytosol of *Amphidinium carterae* by using a specific cell penetrating peptide with the goal to target an eIF4e-1a sequence to inhibit translation. Specific eIF4e knockdown success (up to 42%) has been characterized via microscopy and western blot analysis.

## 1. Introduction

Dinoflagellates are single-celled eukaryotes and members of the Alveolate lineage [1]. Dinoflagellates exhibit extremely diverse trophic strategies, including predation, photo-autotrophy, mixotrophy, and intracellular parasitism [2,3]. Most cultured dinoflagellate species are photosynthetic, making them key marine primary producers. They are well-known for bloom formation in coastal waters, making toxins that bioaccumulate in the food chain, producing bioluminescence, and as coral symbionts [4,5,6].

Climate-change has caused a warming of the Earth’s oceans, benefitting the formation of harmful algal blooms [7]. Of the algal species that have been reported as producing marine harmful blooms, 75% are dinoflagellates [8,9,10,11]. Accumulation of dinoflagellates in coastal waters has begun to increase the presence of red tides, bringing with it fish mass mortality and marine toxin-derived disease in humans [12]. Increasing water temperatures provide optimum growth conditions for many dinoflagellates, allowing for increased toxic effects on their environment [13,14]. Globally, previous research has confirmed the mechanism and structure of some of these toxins [15,16,17,18,19,20].

A hallmark of these toxic blooms can be traced to the production of complex secondary metabolites. Some of these toxins are thought to assist in prey capture through the formation of a nonspecific pore upon complexation with prey’s sterol membrane components [16,21]. Unfortunately, research into the biosynthesis of these dinoflagellate toxins is sorely lacking. This is in large part due to dinoflagellates having unusual cell biology [22]. Their genomes are larger than typical protists, with about 1.2–112 × 10^9^ base pairs of DNA per haploid genome [10], whereas other protist genomes range in the millibases [23,24]. Dinoflagellate chromosomes are condensed into liquid crystalline states throughout the cell cycle and lack nucleosomes, instead using histone-like proteins (HLPs) that are more similar to bacterial DNA binding proteins. Many dinoflagellate genes are organized in multiple copies as tandem repeats, some of which may be present in up to ~10^5^ copies. Increasingly transcriptomic data has shown that dinoflagellates express numerous genes, yet about 50% have no match to known sequences [25]. The function of these sequences, as well as the effects of identified sequences, still need to be established through functional genomic studies.

Control of post-transcriptional regulation in dinoflagellates is currently enigmatic, with mRNA levels showing no correlation to protein production. Because dinoflagellates are believed to regulate at the translational level, rather than during transcription, translation factors are of great interest when understanding dinoflagellate metabolism. In most known eukaryotic translation systems, eIF4Es function as a rate-limiting step toward protein synthesis [26,27]. eIF4E is part of an extended gene family found exclusively in eukaryotes. This translation factor binds to the mRNA cap to recruit the ribosome for translation initiation. In most studied eukaryotic systems (excluding plants), the eIF4E-1 family member is expressed ubiquitously in all cell types from a single copy, such as in *Homo sapiens* or *Saccharomyces cerevisiae* [28,29,30,31,32]. Early studies speculated that eukaryotic systems contain a single gene that encodes eIF4E [33], but since then sequencing projects have revealed that many organisms contain multiple genes encoding proteins that have sequence similarity to the recognized eIF4E [31,34,35,36]. In the case of dinoflagellates, genes regularly appear in multiple copies, with eIF4E being no exception [37,38]. These gene copies commonly appear as slightly different variants with distinctive degrees of diversity. Prior transcriptome analysis of eleven dinoflagellate species has established that each species encodes between eight to fifteen eIF4E family members, a number surpassing that found in any other eukaryotes, including other alveolates [22].

Core dinoflagellate eIF4E translation factors are divided into 3 clades (1, 2, and 3), along with 3 subclades within each (a, b, c); with a total of 9 members. Our previous work has shown that these eIF4E family members display divergences at critical amino acids, suggesting the family members are functionally distinct [22,39]. Of these 3 major clades, eIF4E-1 stands out as the most duplicated, and with the lowest number of substitutions. Based on the expression levels of the subclades, our lab has theorized that eIF4E-1a is likely the primary translation initiation factor.

Although expression of subclade eIF4E-1a is highest of all the family members, dinoflagellates still generate a greater diversity and degree of eIF4E duplications than seen in other eukaryotes [22]. Understanding their various roles will bring us closer to understanding how dinoflagellates adapt to their environment, giving insight into harmful algal bloom formations, as well as their production of complex secondary metabolites and toxin biosynthesis. Generally, eIF4E family members are known to have different roles in metazoan gene expression [40]. Similarly, we predict that dinoflagellate eIF4Es will have distinct functions, allowing for an increased dependence on the translational control of gene expression [22]. Determining the role of the eIF4E family members requires a method of knocking down their expression. The unusual cell biology of dinoflagellates makes common gene knockout strategies impractical, limiting the amount of genetic research that can be applied. This is because gene knockouts require a “deletion” of all operable gene-copies, which can be difficult to obtain when many copies with slight variations exist [38]. In this case, gene knockdown strategies are the most feasible, as they target the functional transcripts produced by the gene copies, which in many cases remain less diverse [39,41].

In particular, we are pursuing the use of an antisense-based knockdown approach in order to study how a decrease in target gene expression effects dinoflagellate metabolism. In prior research, the introduction of antisense-oligomers to dinoflagellate cells has been hampered by their thick, cellulosic cell wall [42,43]. Other studies have bypassed this obstacle by preparing spheroplasts, cells with a completely or partially removed cell-wall, beforehand [42,43,44]. Spheroplast production is done by incubating cells on plates in a polyethylene glycol (PEG) solution, which promotes fusion of the vesicles and cell membrane, and ultimately a decrease in total cellulose. So far only two studies have been successful in achieving gene knockdown with this spheroplast procedure; targeting a condensin subunit and targeting a cellulose synthase [42,43,44]. Once introduced, the antisense-oligo was able to bind to cytoplasmic mRNA and knockdown expression of the target gene. Although gene expression could be quantified in this way, it appears that some physiological effects were hidden by the effects of PEG on the cell wall, which causes the cells to lose rigidity. Also, the need for cell plating, rather than cell culturing, immensely limits the species of dinoflagellates that can be studied since many will not grow outside of a liquid medium.

There has also been evidence of RNA interference (RNAi) machinery within dinoflagellates, a naturally occurring mechanism for gene silencing through various methods such as RNA degradation, transcriptional repression and translation inhibition [25,45,46]. One study observed the effects of RNAi silencing tool on the proton-pump rhodopsin and CO_2_-fixing enzyme Rubisco encoding genes in dinoflagellates by introducing small interfering RNAs (siRNAs) to dinoflagellate cultures via immersion. Results showed success in gene suppression within the two dinoflagellate species studied, *Prorocentrum donghaiense* and *Karlodinium veneficum* [25]. This decrease in gene expression was observed with a decrease on overall growth rate for both species as well, compared to the control green fluorescent protein (GFP) labelled siRNA. This knockdown method in dinoflagellates was initially challenging due to the large copy number of target DNA and permanently-condensed chromatin, but recent research has shown that there is a strong possibility that knockdown procedures can be successful.

Recently our team has begun to develop a system for introducing antisense morpholinos into dinoflagellate cells without the use of PEG to warp the cell wall, or the use of RNAi. Instead, we are using a novel delivery peptide that delivers substances via an endocytosis-mediated process that avoids damaging the plasma membrane of the cell [47]. Not only this, but the knockdown process can be done completely by immersion. The peptide and antisense morpholino are added directly to the culture to stimulate endocytosis and morpholino uptake. To test this system on dinoflagellates, we used the manufacturer recommended concentrations of delivery-peptide and antisense-morpholino on a dense culture of *Amphidinium carterae*, a known algal bloom species. The antisense morpholino is targeted to what we believed to be the main dinoflagellate translation factor, eIF4E-1a [22,27,40,48]. Our preliminary data has shown that unlike with PEG addition, the delivery peptide does not cause the cell population to drastically decrease.

## 2. Materials and Methods

### 2.1. Cell Culturing

*Amphidinium carterae (Hulbert)* strain CCMP1314 was grown in ESAW artificial marine media with a salinity of 32 ppt supplemented with f/2 nutrients without silicates at 25 °C [49]. The medium was buffered with 1mM HEPES (pH 8.0). Since bacterized cultures have shown to affect analyses of translation rate in *A. carterae*, the cultures were maintained axenically with an antibiotic solution of kanamycin (50 μg/mL), carbenicillin (100 μg/mL), and spectinomycin (50 μg/mL) [50]. The cultures were grown under 100 μmol/m^2^ s^−^ light. Delivery of the morpholinos (described below) requires constant swirling to keep the reagents in solution, therefore the cultures were also placed on an orbital shaker at 60 rpm [47]. The cultures were allowed to acclimate to the swirling for a week before knockdown reagents were added.

### 2.2. Morpholino Customization and Delivery

The sequence of the initiation factor eIF4E-1a was found by our lab previously [22]. Morpholino antisense oligonucleotides (MOs) are nucleic acid analogues in which DNA bases are bound to a non-charged backbone (morpholine rings linked by phosphorodiamidate bonds) [51]. For our purposes, a translation-blocking MO was created that covered the eIF4E-1a translational start site (5′-TCATTGAAGCTCAAACAAGCCATTG-3′). Specificity for the intended target sites was verified by BLAST analysis against the *Amphidinium carterae* transcriptome. MOs were purchased from GeneTools (Philomath, OR, USA) and modified with a red-emitting fluorescent 3′ Lissamine addition and then used at a concentration of 1 μM and 10 μM. Standard control MOs with the Lissamine addition were ordered as well (5’-CCTCTTACCTCAGTTACAATTTATA-3’).

MOs were delivered using Endo-Porter reagent (GeneTools, Philomath, OR, USA) at a concentration of 4 µM. Cultures of *A. carterae* seeded in 12-well plates were treated with either Endo-Porter, MO, or both. Three biological replicates of each treatment were performed.

### 2.3. Cell Counts and Fluorescence Quantification

Following the addition of 4 µM Endo-Porter and 1 µM or 10 µM MO, the viability of cultures of *A. carterae* were observed over a 96-h period by measuring their autofluorescence via flow cytometry. Measurements of the cultures within the first hour of treatment were labelled as Hour 0. Cell counts for each condition were determined on a BD C6 Accuri Flow Cytometer (BD Biosciences, San Jose, CA, USA), equipped with laser excitation at 488 and 640 nm and emission at 533/30, 585/40, and >670 nm. The FSC-A and fluorescence channels were used to select for live cells; from this selection, cells with Lissamine emission signals were detected (585 nm). Cells were grouped into low and high Lissamine fluorescence, with high fluorescence showing an intensity minimum of 10^4^ relative fluorescence units (RFUs).

### 2.4. Cell Imaging

Images of A. carterae cells with and without MO treatment were taken on a STELLARIS confocal microscope (Leica Microsystems, South San Francisco, CA, USA), equipped with 405, 552 and 638 nm lasers, and PMT and HyD detectors collecting emission within 590–600 nm and 680–720 nm, respectively.

### 2.5. Quantification of Protein Expression

Protein expression was quantified by Western blotting. For the initial Western blot analyses done for the 1 μM concentration of morpholinos, the cell density of the cultures were quantified via flow-cytometry to create equal volume pellets containing ~75,000 cells for each condition 48 h post-treatment. A NuPAGE 4 to 12% Bis-Tris 1.5 mm Mini Protein Gels was used. Cell pellets were prepared for electrophoresis in 3× sample buffer (Blue Loading Buffer Pack, New England Biolabs, Ipswich, MA, USA), heated to 96 °C for 10 min and centrifuged for 2 min at 10,000× *g*; from the 15 μL total, 10 μL of each extract was electrophoresed at 165 V until the dye front reached the bottom of the gel. The gel was transferred to a membrane with the Trans Blot Turbo Transfer system. Protein loading and relative expression levels were verified by probing the same blot with anti-eIF4E-1a mouse monoclonal and HRP-conjugated anti-mouse IgG. The labeled bands, as well as band intensities, were detected with ImageLab software.

For the subsequent experiment testing the higher 10 μM concentration of morpholinos, equal quantities of whole-cell lysates containing 100,000 cells were prepared from each of the triplicate sample cultures at 48 and 96 h. Samples were once again split equally over two NuPAGE 4–12% Bis-Tris Gels and run for 50 min at 165 V. Both gels were transferred to a membrane with the Trans Blot Turbo Transfer system. To control for cell loading error, total protein was detected on one membrane using No-Stain Protein labelling Reagent (Invitrogen, Waltham, MA, USA). Summation of total protein in each lane was found using ImageJ. Protein loading and relative expression levels were again verified by probing the second blot with anti-eIF4E-1a mouse monoclonal and HRP-conjugated anti-mouse IgG. The labeled bands were detected with ImageLab software, as well as band intensities. The relative production of eIF4E-1a was analyzed by dividing the eIF4E-1a volume by the summation of total protein.

### 2.6. Statistical Considerations

All conditions were performed in triplicate. Pairwise sample comparisons within timepoints were analyzed in R-Studio with *t*-tests using pooled standard deviations [52]. A *p*-value of <0.05 was considered statistically significant.

## 3. Results

### 3.1. Cell Viability

Immediately post-treatment, the cultures displayed a suspension of growth, but recovery was observed at 24 h (Figure 1). The cultures containing the antisense morpholino also showed significant differences in growth compared to the control immediately after treatment addition and at both 24 and 48 h.

### 3.2. Uptake of the Morpholino

The intensity of the Lissamine signal within cells was measured by flow cytometry (Figure 2). Peak Lissamine fluorescence was observed at 48 h, with about 13% of the population uptaking a high amount of fluorescently-tagged morpholinos [47], and waned after this timepoint. The mean Lissamine fluorescence per cell with the Endo-Porter delivery system plus the MO was over 50X greater than that of the control, and the median was over 1.5X greater, showing significant uptake of the MO in *A. carterae*, as well as a large positively-skewed distribution of uptake efficiency (Appendix A). Cells were also able to uptake morpholino without Endo Porter, but to a significantly lower degree, with no correlation to experimental duration.

Images of cells were also captured by confocal microscopy (Figure 3). Lissamine fluorescence was detected within the range of 590–600 nm (peak 593 nm), and autofluorescence was measured between 700–720 nm. Confocal images display a diffuse pseudo-blue coloring within the cytoplasm of the dinoflagellate cells, as well as in a large area near the nucleus, indicating both diffuse and localized morpholino presence (Figure 3).

### 3.3. Initial Western Blot Analysis

From the initial Western Blot analysis, we found a statistically significant decrease in the expression of eIF4E-1a of about 30% compared to the control in cultures containing both the 1 μM morpholino and Endo Porter (Figure 4). Cultures with Endo Porter showed a decrease of 11% in eIF4E-1a expression over cultures without Endo Porter, although the difference was not statistically significant.

### 3.4. Increase in MO Concentration

Once the concentration of morpholinos was increased from 1 μM to 10 μM, we found that with the custom eIF4E-1a target morpholino and Endo Porter there was a decrease in population eIF4E-1a expression at 48 h compared to the control of about 42% (Figure 5). This was an increase from the 30% found when using only 1 μM of MO, constituting a total 11.8% decrease with the increased MO concentration (Table 1). The cultures containing eIF4E-1a-target morpholino only also showed reduced target-protein production compared to the control of 21%. When compared to the culture containing only the standard non-target morpholino and Endo Porter, both of the cultures containing the eIF4E-1a-target morpholino with and without the Endo Porter appeared to exhibit a significant decrease of 47% and 29%, respectively (Figure 5). No significant difference was found between the cultures containing eIF4E-1a-target morpholino with or without the Endo Porter, although the cultures with the Endo Porter showed an average decrease of 25% compared to the cultures without.

Interestingly, we also found that expression levels became relatively similar after 96 h, showing the temporary effects of the morpholino (Appendix A).

## 4. Discussion

Here we have shown that novel delivery peptide technology has allowed for successful introduction of custom antisense morpholinos into *A. carterae* cells without significantly decreasing viability. This method delivers substances into the cytosol of cells by an endocytosis-mediated process that avoids damaging the plasma membrane of the cell [47,53]. With this new system, exploration into the key players of different metabolic pathways may be closer than expected for dinoflagellates.

The above data was collected using the manufacturer recommended concentrations of reagents, but as we look further into these systems, we hope to optimize the outcome by adjusting protocol values. Currently, with 1 µM of MO added, we have been able to successfully introduce a high concentration of the MO into approximately 13% of the population (Figure 2), with a significant decrease in eIF4E-1a protein production. Dose-dependent effects of the MOs have also been observed, with an increase in MO concentration from 1 μM to 10 μM resulting in a decrease in eIF4E-1a from 30% to 42%; an 11.83% decrease in total (Table 1). Optimistically we would like to increase this percentage to a level where protein production is functionally suspended, or be able to separate cells from the population based on their MO uptake.

Interestingly. western blot analyses for both the initial 1 μM and subsequent 10 μM MO experiment showed no statistically significant difference between the cultures with and without Endo-Porter, although the cultures with the Endo-Porter showed consistently lower average eIF4E-1a protein produced (Figure 4 and Figure 5). This would indicate that even without a delivery peptide, *A. carterae* cells are able to uptake the MO. Reasons for MO uptake without a delivery peptide are still unclear. Over a million identified peptide sequences within various dinoflagellates are still of unknown function and origin, making their evolutionary history ambiguous [54]. Dinoflagellates are thought to have undergone multiple organellogensis events, where the genome of endosymbiotic algae becomes a plastid and/or genes from the endosymbiont are transferred to the nucleus [55]. The evolutionary nature of dinoflagellates to accept foreign genes appears to be high [56]. One theory for why the *A. carterae* cells took up the unaided MO could be the possibility that dinoflagellate systems are more open to horizontal gene transfer (HGT) than previously imagined [57]. Recent studies have shown that genomes within dinoflagellates may be more open to foreign contributions from both bacteria and eukaryotes compared to other organisms [57,58]. Further research needs to be conducted to follow how foreign genetic material is received by dinoflagellates, as well as how transcripts are processed.

A wide distribution of MO uptake may account for the discrepancy of measured eIF4E-1a protein production compared to the cellular MO uptake (Appendix A). Although we suggest about 13% of the cells have a high uptake after addition of 1 μM of MO, the eIF4E-1a production was decreased by about 30% according to the western results (Figure 4). We assume that there is a range of efficiency uptake within the population, so some cells may be producing little or no eIF4E-1a, while others may be producing their average amounts. Once again, changes to the concentrations used or the use of a cell-sorter may be necessary to observe higher gene knockdown efficiency.

MO localization does occur within the *A. carterae* cells, usually in a large area by the nucleus (Figure 3). In previous studies, Endo-Porter sometimes results in unsuccessful endosomal acidification, and therefore no release of the MO into the cytoplasm from their vesicle, but these localization points are usually known to appear as punctate fluorescence throughout the cell [47]. The singular, large area of localization may signify MO aggregation in the nucleolus or an RNA-granule [55,59,60,61,62]. Dinoflagellates are known to have very unusual nuclei, specially named the “dinokarya.” Among other peculiar features are recently discovered “nuclear tunnels” which extend from the nuclear envelope of *Polykrikos kofoidii*, specifically during mitosis [59]. These nuclear envelope tunnels are also connected with a membranous structure throughout the nucleus known as the “nuclear net”. The discovery of these structures adds a new level of complexity to the dinoflagellate nuclear membrane, and may allude to more complex processing for transcription and translation. The unusual nuclear envelope tunnels along with our MO localization could very well be connected. Since dinoflagellates are known to regulate gene expression at the translational level, this cellular organization of mRNA may be a crucial step for gene expression, which needs to be analyzed further.

In theory, this method of gene knockdown is progress towards expanding research into the biosynthetic pathways within dinoflagellate. Currently, this research is sorely lacking, largely due to their complex genomes and unusual cell biology. Translation regulation is now understood as a crucial step in gene expression, far beyond that of transcriptional control [22,48]. In addition, further understanding of the components of the translation machinery is required to understand the expression of specific genes, which make up the dinoflagellate “translational toolkit”. As we work to optimize this procedure, other pathways could be targeted as well in other species of dinoflagellates.

One interesting area would be to target the transcript responsible for toxin production. The function of toxins produced by dinoflagellates has been theorized but is still unknown. Theories include allelopathy, prey-capture, and as a defense [18]. One way to discern the function of the toxins would be to knockdown their expression to see how this affects feeding and swimming behavior. Mixotrophic characteristics of dinoflagellate species known to produce toxins could be monitored through photosynthesis and respiration rates, as well as swimming behavior via digital holographic microscopy [63]. This data could produce more evidence to the intended functionality of dinoflagellate toxins.

## 5. Conclusions

This primary study provides proof-of-principle for the possibility to specifically down-regulate gene expression in dinoflagellates using antisense morpholinos and a novel delivery system. Additional work is necessary to validate and optimize these findings, and to extend them to other biosynthetic pathways. Further work will also be required to investigate the effects of gene knockdown on the various eIF4E family members in order to possibly unveil a translation toolkit used by dinoflagellates to regulate gene expression. Studies involving toxin biosynthesis pathways would also benefit from successful knockdowns in order to identify functionality and key synthesis steps.

## Figures and Tables

**Figure 1 microorganisms-10-01131-f001:**
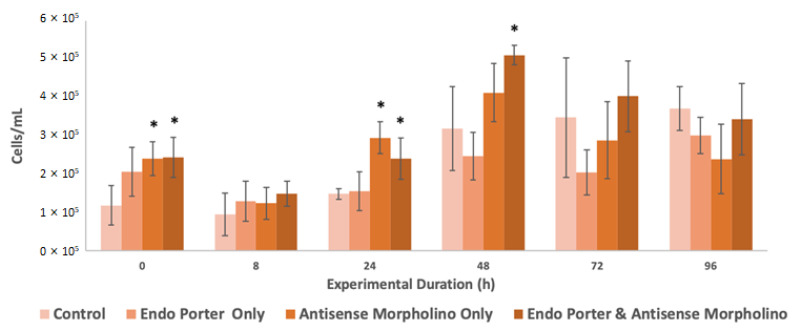
Flow cytometer population counts from *Amphidinium carterae* cultures (N = 3). Conditions included the control, Endo-Porter only, antisense morpholino only, and the Endo-Porter and antisense morpholino combined. Statistical analysis was done using *t*-tests with pooled standard deviations. * Significantly different from ‘Control’ based on a *p*-value of < 0.05.

**Figure 2 microorganisms-10-01131-f002:**
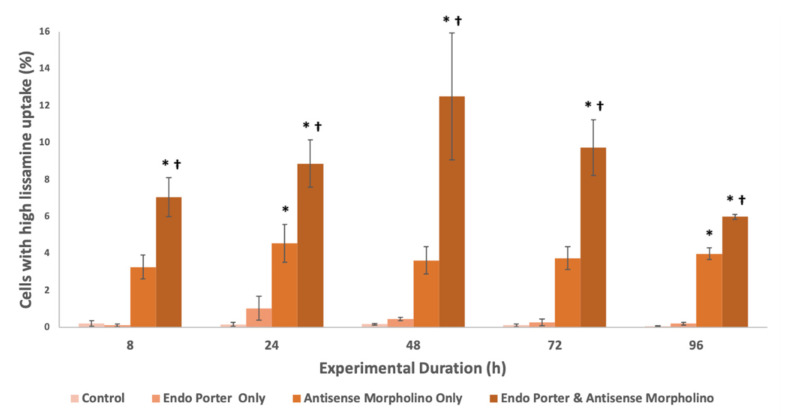
Percent of A.carterae populations with high Lissamine uptake after 48 h (cutoff 104 RFUs). Conditions included the control, Endo Porter only, antisense morpholino only, and the Endo Porter and antisense morpholino combined (N = 3). Statistical analysis was done using *t*-tests with pooled standard deviations. * Significantly different from ‘Control’ based on a *p*-value of < 0.05. † Significant difference between “Antisense Morpholino Only” and “Endo Porter & Anti-sense Morpholino” based on a *p*-value of < 0.05.

**Figure 3 microorganisms-10-01131-f003:**
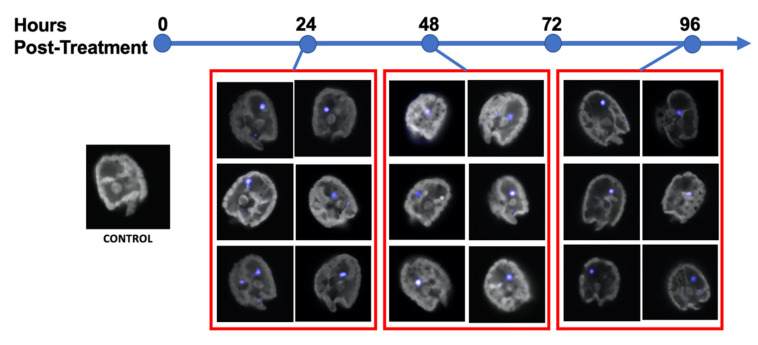
Confocal images of *Amphidinium carterae*. A control cell with no antisense morpholino introduced is on the far left. Red boxes encompass images of cells from a culture with Endo Porter and antisense-morpholino, fluorescently tagged with Lissamine (593 nm, pseudo-blue) at 24-, 48- and 96-h post-treatment. The grey areas are the auto-fluorescence emitted within 720–759 nm.

**Figure 4 microorganisms-10-01131-f004:**
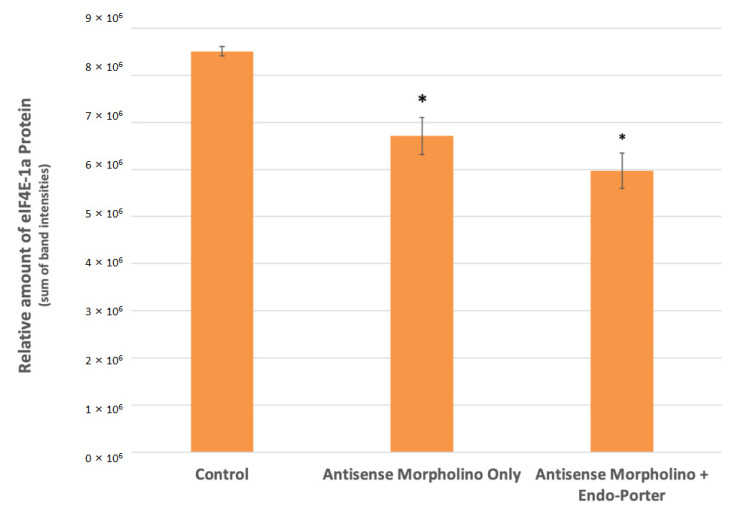
Western blot analyses for eIF4E-1a concentrations within control and treated cells at 48 h post-MO [1 μM] addition (N = 3). Protein loading and relative expression levels was verified by probing with anti-eIF4E-1a mouse monoclonal and HRP-conjugated an-ti-mouse IgG. eIF4E-1a Protein area volumes were reduced by 30% after introduction of custom translation blocking morpholino and Endo Porter after 48 h. Statistical analy-sis was done using *t*-tests with pooled standard deviations. * Significantly different from “Control” based on a *p*-value of < 0.05. No significant difference found between “Antisense Morpholino Only” and “Antisense Morpholino + Endo Porter” based on a *p*-value of < 0.05.

**Figure 5 microorganisms-10-01131-f005:**
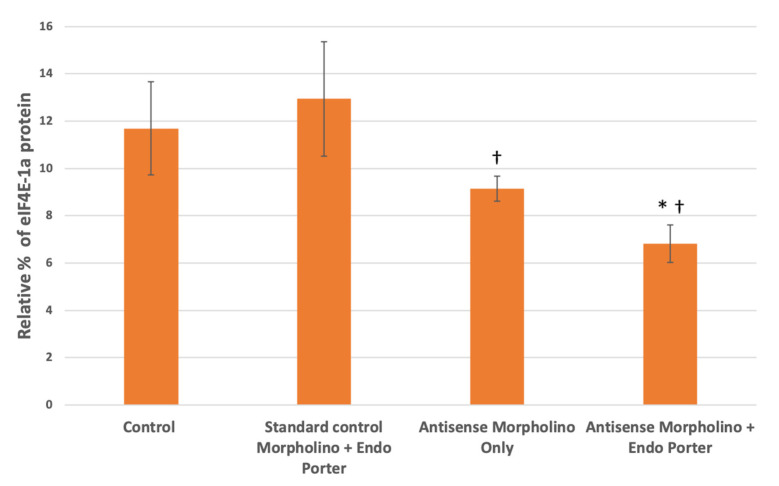
Western blot analyses for eIF4E-1a concentrations within control and treated cells at 48 post-MO [10 μM] addition (N = 3). Protein loading and relative expression levels was veri-fied by probing with anti-eIF4E-1a mouse monoclonal and HRP-conjugated anti-mouse IgG and compared to total protein volumes. Relative eIF4E-1a levels are lower in Am-phidinium population after 48 h of being subjected to custom translation-blocking morpholinos. Statistical analysis was done using *t*-tests with pooled standard deviations. * Significantly different from ‘Control’ based on a *p*-value of < 0.05. † Significantly different from “Standard Morpholino + Endo Porter” based on a *p*-value of < 0.05. No significant difference found between “Antisense Morpholino Only” and “Antisense Morpholino + Endo Porter” based on a *p*-value of < 0.05.

**Table 1 microorganisms-10-01131-t001:** Volume of eIF4E-1a produced compared to the control. Volumes of eIF4E-1a were quantified via western blot and compared to the density of a whole protein stain per sample (N = 3). Percent of eIF4E-1a for each sample were compared to the control sample to describe relative eIF4E-1a production. Statistical analysis was done using *t*-tests with pooled standard deviations.

Morpholino Concentration	1 μM	10 μM	
	Relative eIF4E-1a Production		Percent Change
Antisense Morpholino Only	78.8% *	78.3%	0.5%
Antisense Morpholino + Endo Porter	70.1% *	58.3% *	11.8%

***** Significant difference when compared to the control based on a *p*-value of **<** 0.05.

## Data Availability

Not applicable.

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
