# Peer review of "A Strategy for Gene Knockdown in Dinoflagellates"

_microorganisms, 2022, doi:10.3390/microorganisms10061131_

Round 1
Reviewer 1 Report
General comments:
This manuscript describes the use of a novel technique in dinoflagellates to enable gene knockdown, including an interesting delivery method. Due to the unusual nature of dinoflagellate cells common methodologies in cell biology are difficult and this study provides an interesting and promising approach for the field. Overall, the manuscript is well-written and clear (although the introduction could be improved). Some specific comments are listed below.
Specific comments:
- “Of the algal species that have been reported as producing harmful blooms, 75% are dinoflagellates”. Do you mean marine harmful algal blooms here?
- “Accumulation of dinoflagellates in coastal waters has begun to increase the presence of red tides, bringing with it fish mass mortality and marine toxin-derived disease in humans (Pagliara & Caroppo, 2012; Pagliara & Caroppo, 2012). As climate-change affects the Earth’s oceans, the water can warm, benefitting the formation of harmful algal blooms (Aquino-Cruz & Okolodkov, 2016). Increasing water temperatures provide optimum growth conditions for many dinoflagellates, allowing for increased toxic effects on their environment (Brownlee et al., 2008; Galimany et al., 2008).” This paragraph is not well-written, please edit. Some more up to date literature would be useful as well.
- “Our group’s previous research has confirmed the mechanism and structure of some of these toxins (Peng et al., 2010).” I think this sentence underrepresents the wider international research in this space. Please include other key references.
- Figure 2: Please be consistent with your use of delivery peptide and Endo Porter.
- Line 244: “Images of treated cells show evidence of both diffuse and localized morpholino presence.” This sentence is very brief and doesn’t provide much information on the results seen. Please expand on the results of the confocal images. For example, this sentence in the discussion (Line 315) would be more appropriate in the results. “MO localization does occur within the A. carterae cells, usually in a large area by the nucleus (Figure 3).”
- Throughout the manuscript are small typos and formatting mistakes, please proof-read carefully.
Author Response
The authors would like to thank you for your thoughtful and thorough review of the attached manuscript. The comments made have been considered and addressed. Below are the responses to each point:
- “Of the algal species that have been reported as producing harmful blooms, 75% are dinoflagellates”. Do you mean marine harmful algal blooms here?
Yes, marine. This modifier has been added to the sentence.
- “Accumulation of dinoflagellates in coastal waters has begun to increase the presence of red tides, bringing with it fish mass mortality and marine toxin-derived disease in humans (Pagliara & Caroppo, 2012; Pagliara & Caroppo, 2012). As climate-change affects the Earth’s oceans, the water can warm, benefitting the formation of harmful algal blooms (Aquino-Cruz & Okolodkov, 2016). Increasing water temperatures provide optimum growth conditions for many dinoflagellates, allowing for increased toxic effects on their environment (Brownlee et al., 2008; Galimany et al., 2008).” This paragraph is not well-written, please edit. Some more up to date literature would be useful as well.
The paragraph has been restructured to be more comprehensible. More current literature has been cited as well.
- “Our group’s previous research has confirmed the mechanism and structure of some of these toxins (Peng et al., 2010).” I think this sentence underrepresents the wider international research in this space. Please include other key references.
Other references on toxin mechanisms have been added.
- Figure 2: Please be consistent with your use of delivery peptide and Endo Porter.
Legend has been changed to “Endo-Porter” consistently
- Line 244: “Images of treated cells show evidence of both diffuse and localized morpholino presence.” This sentence is very brief and doesn’t provide much information on the results seen. Please expand on the results of the confocal images. For example, this sentence in the discussion (Line 315) would be more appropriate in the results. “MO localization does occur within the A. carterae cells, usually in a large area by the nucleus (Figure 3).”
The sentence has been expanded upon to better describe the location of morpholino presence.
- Throughout the manuscript are small typos and formatting mistakes, please proofread carefully.
The manuscript has been proofread again multiple times, and small typos and formatting mistakes have been identified and fixed.
Reviewer 2 Report
This is a very well-written article explaining the principle and methodology to regulate the gene expression in dinoflagellates using morpholinos and a novel peptide delivery system. Since this is a preliminary research, the authors have done good work on aptly explaining each concept in the introduction which gives a better understanding to the reader. The results presented in this article show that by utilizing this technology, the authors were able to significantly downregulate the protein expression of eIF4E-1a. Although further work needs to be done on investigating the downstream effects of eIF4E family knockout, it would be more interesting to know the effect it has on toxin production, if any, and whether this could be utilized to alter the toxin biosynthesis. A few minor corrections required are
- Line 91: ...will bring us closer...
- Line 253 and 266: .. decrease in population of eIF4E-1a expression.. The word population could be replaced as protein expression. It appears to refer to the dinoflagellate population.
- Please check the spacing between number and units throughout the document. eg: 1mm vs 1 mm
Author Response
The authors would like to thank you for your thoughtful and thorough review of the attached manuscript. The comments made have been considered and addressed. Below are the responses to each point:
- Line 91: ...will bring us closer...
The error has been fixed.
- Line 253 and 266: .. decrease in population of eIF4E-1a expression.. The word population could be replaced as protein expression. It appears to refer to the dinoflagellate population.
The word population has been replaced with expression.
- Please check the spacing between number and units throughout the document. eg: 1mm vs 1 mm
Number/units have been identified and given a space between them.
Reviewer 3 Report
The paper A Strategy For Gene Knockdown In Dinoflagellates is a crucial paper on the development of a new peptide-based vehicle capable of going through cell membranes and deliver a morpholino to inhibit specific mRNA. Even though the technique's success is not complete yet, the development of this particular technique is of great importance for the study of dinoflagellates' unusual nuclei.
The paper is thorough and well written; these findings should be published very soon. Congratulations to the authors.
Nevertheless, some details should be taken care of.
The Introduction contains all the essential information presented attractively. It was not only interesting but fascinating to read it.
Materials and Methods:
Line 132: please explain GPF previous to using the abbreviation.
L152: salinity has no units.
L152: did you use f/2 nutrients modified without silicates? If so, please mention it.
L153: Why use axenic cultures? If you remove their microbiome, does this have consequences on their transcriptome? Can't this be done in normal xenic (bacterized) cultures?
L156: There are many reports of Amphidinium not growing in shakers, why did you shake them? Please explain.
L190: Please specify how you counted these 75,000 cells.
Results and discussion
L 219-221: there is a significant difference in cells mL-1 since day 0 (setup of the experiment). Results showing significant differences on the other days are unreliable since this happened from the beginning.
L261: How can you explain a lack of significant difference between "antisense morpholino only" and "antisense morpholino + endo porter"? This issue is not discussed, and it is an important one.
L 268: Supp Table 2 should not be supplementary material. It is easier to understand the table than the explanation written in 3.4.
The reference list must be revised and improved. There are several errors.
Overall details: the authors might think that correct writing is unimportant when the information is crucial. However, there are too many typing and writing errors (marked in red in the attached documents), such as a lack of spaces between numbers and units (see examples L166, L193, and so on). The lack of correct use of standard abbreviations such as h (hour), min (minutes), and s (seconds).
The graphs and figures should not have outer margins or inner titles. Information should be complete in the figure caption. I recommend using different colors for different treatments instead of different tones of the same color.
Please check the attached document carefully.
Author Response
The authors would like to thank you for your thoughtful and thorough review of the attached manuscript. The comments made have been considered and addressed. Below are the responses to each point:
Materials and Methods:
- Line 132: please explain GPF previous to using the abbreviation.
An unabbreviated name has been added.
- L152: salinity has no units.
Units of parts per thousand (ppt) has been added.
- L152: did you use f/2 nutrients modified without silicates? If so, please mention it.
The exclusion of the silicates has been specified.
- L153: Why use axenic cultures? If you remove their microbiome, does this have consequences on their transcriptome? Can't this be done in normal xenic (bacterized) cultures?
This is a great point. Currently, we are working with axenic cultures because we are hoping to continue this research by evaluating translation rate with 35S-methionine incorporation. In the Liu et al. 2017 paper cited, it was discovered that the addition of a eukaryotic-specific inhibitor of protein synthesis, cycloheximide, had little effect on the culture’s 35S-methionine incorporation when bacterized. They also found with the addition of a bacterial-specific protein synthesis inhibitor, 35S-methionine incorporation was significantly reduced. So for our future studies, axenic cultures will be useful.
The use of morpholinos for bacterized cultures would be interesting, and also another great direction for future research.
- L156: There are many reports of Amphidinium not growing in shakers, why did you shake them? Please explain.
The delivery peptide, Endo-Porter, requires constant swirling to stay in solution. We set the cultures to be swirled at the minimum speed, and allowed the populations to acclimate over a week before adding the knockdown reagents. This did not seem to significantly affect the viability of the cultures.
- L190: Please specify how you counted these 75,000 cells.
We used flow cytometry to count the 75000 cells. This has been added to the Methods.
Results and discussion
- L 219-221: there is a significant difference in cells mL-1 since day 0 (setup of the experiment). Results showing significant differences on the other days are unreliable since this happened from the beginning.
Yes, population densities were variable among samples. Figure 1 is meant to show general trends in cell viability with the different treatments. All subsequent conclusions are drawn with the culture densities in mind, such as the percent of cells with morpholino uptake in Figure 2 (percent fluorescent cells within the population).
- L261: How can you explain a lack of significant difference between "antisense morpholino only" and "antisense morpholino + endo porter"? This issue is not discussed, and it is an important one.
Thank you for pointing this out. A section has been added to the Discussion to suggest theories for this lack of statistical significance.
- L 268: Supp Table 2 should not be supplementary material. It is easier to understand the table than the explanation written in 3.4.
The material has been changed to a primary table.
- The reference list must be revised and improved. There are several errors.
The reference list has been revised.
- Overall details: the authors might think that correct writing is unimportant when the information is crucial. However, there are too many typing and writing errors (marked in red in the attached documents), such as a lack of spaces between numbers and units (see examples L166, L193, and so on). The lack of correct use of standard abbreviations such as h (hour), min (minutes), and s (seconds).
The manuscript has been revised and typos/grammatical errors have been fixed. Spacing has also been added between numbers and units.
- The graphs and figures should not have outer margins or inner titles. Information should be complete in the figure caption. I recommend using different colors for different treatments instead of different tones of the same color.
Figure formatting has been changed to not include margins or inner titles. The monochromatic colors have been kept to allow better visibility for color vision deficient readers.
- Please check the attached document carefully.
We have reviewed the attached document and noted your highlighted regions. Thank you for the careful review.
Reviewer 4 Report
On account of the incomplete data concerning dinoflagellate IF4E expression which generate great diversity and degree of eIF4E duplication taking such a topic of examinations up is justified. The paper is relevant to general molecular process connected with the deepening of knowledge how dinoflagellates create quickly harmful bloom and produce complex of secondary metabolites and toxins. Authors presented very interesting results which are broadening the information in this scope.
The title enough reflects the content of the paper. Abstract and Key words are appropriate informative and adequate to the paper’s content.
The paper substantially is performing all requirements which are being put for examinations of this type.
The research hypothesis put in the introduction is correct and with clear argumentation. The construction of figures and tables do not create objections. All tables and figures include the main results that have been used in the article. They are graphically very readable.
The presentation of Results is logical and put in order so without the problem it can be watch progress of research data.
Discussion is complete, and it is establishing at put earlier aims of the work.
References used in the article were correctly selected to presented problems.
The manuscript has been prepared very carefully.
In my opinion, this manuscript can be accepted in present form for publication in “Microorganisms”.
Author Response
The authors would like to thank you for your thoughtful and thorough review of the attached manuscript. We appreciate the enthusiastic comments.